# Black Chokeberry (*Aronia melanocarpa*) Functional Beverages Increase HDL-Cholesterol Levels in Aging Rats

**DOI:** 10.3390/foods10071641

**Published:** 2021-07-15

**Authors:** Elena Daskalova, Slavi Delchev, Lyudmila Vladimirova-Kitova, Spas Kitov, Petko Denev

**Affiliations:** 1Department of Anatomy, Histology and Embryology, Medical University, 4000 Plovdiv, Bulgaria; sldel@abv.bg; 2I-st Department of Internal Diseases, Section of Cardiology, Medical University, 4000 Plovdiv, Bulgaria; kitov@vip.bg; 3Clinic of Cardiology, St. George University Hospital, 4000 Plovdiv, Bulgaria; kitovspas@yahoo.com; 4Laboratory of Biologically Active Substances, Institute of Organic Chemistry with Centre of Phytochemistry, Bulgarian Academy of Science, 4000 Plovdiv, Bulgaria

**Keywords:** black chokeberry (*Aronia melanocarpa*), aging, lipids, HDL-cholesterol

## Abstract

Plant-based foods rich in phenolic phytochemicals are among the promising strategies to counteract age-related changes in lipid profile. *Aronia melanocarpa* (AM) fruits are a rich source of phenolic compounds possessing lipid-modulating effects. The present study investigated the effect of 3-month supplementation of AM-based functional beverages on the lipid profile of healthy aging rats. Male Wistar rats (*n* = 40) were separated into five groups: (YC) young controls (2-month-old); (AC) adult controls (13-month-old); (A) adult animals supplemented with pure AM extract; (A + P) adult animals supplemented with pectin-enriched (1%) AM extract; (A + H) adult animals supplemented with AM extract enriched with a herbal mixture. Total cholesterol (TC), triglycerides (TG), HDL-cholesterol (HDL-C), LDL-cholesterol (LDL-C) and atherogenic indices were investigated at the end of the study. Adult controls demonstrated age-related dyslipidemia resulting in decreased HDL-C, and increased TG and TC/HDL index. The supplemented groups showed a significant increase in HDL-C levels: A + P (1.49 mmol/L) and A + H (1.61 mmol/L), respectively, vs. AC (1.09 mmol/L), *p* < 0.05. The TC/HDL-C and LDL-C/HDL-C indices were decreased in the A + P and A + H groups in comparison to the AC group (*p* < 0.05). These results indicate that supplementation with polyphenol-rich AM beverages can successfully alter HDL-C levels and this effect is further potentiated by pectin and herbs.

## 1. Introduction

Age and sex are physiological factors that strongly affect plasma lipid levels in a number of species. Dyslipidemia is a metabolic alteration characterized by elevated fasting blood levels of total cholesterol (TC), low-density lipoprotein cholesterol (LDL-C), triglycerides (TG), and decreased levels of high-density lipoprotein cholesterol (HDL-C). Dyslipidemia is known to be a risk factor for developing insulin resistance, endothelial dysfunction, hypertension and, most of all, cardiovascular disease (CVD) [1]. Furthermore, dyslipidemia is a proven independent risk factor for vascular pathology because even when other risk factors are absent, atherosclerosis progresses. The latter is considered to be the cause of more than half of the cases of ischemic heart disease in the world [2]. HDL-C is one of the key factors determining cardiovascular risk. The protective role of HDL-C was determined for the first time in an epidemiologic study in 1970, when Framingham convincingly proved that HDL-C is the strongest predictor of developing ischemic heart disease in men and women over 49 years of age. Recent evidence has proven that a very small change in HDL-C—around 0.26 mmol/L—leads to a reduction in risk by 2% in men and 3% in women [3,4].

Low HDL-C levels present an unresolved treatment problem. It has been found that lifestyle changes increase HDL-C, and many of the mechanisms underlying its occurrence have been revealed. The latter include physical activity—5–10% (increased lipoprotein lipase, pre-β-HDL, activated reverse cholesterol transport); cessation of smoking—5–10% (raised lecithin-cholesterol acyltransferase, activated reverse cholesterol transport, inhibited cholesterol ester transfer protein); weight loss—5–20% (increased lecithin-cholesterol acyltransferase and lipoprotein lipase, activated reverse cholesterol transport); moderate alcohol consumption—5–15% (increased member 1 of human transporter sub-family ABCA, apolipoprotein A-1, paraoxonase, reduced cholesterol ester transfer protein); Mediterranean diet/unsaturated fatty acids—0–5% (increased atheroprotective lipoproteins). The only group of drugs that modulate HDL-C levels with a proven clinical benefit in lowering cardiovascular incidents is HMG-coenzyme A reductase inhibitors (statins). Among all statins, only two are of practical significance—Rosuvastatin (increases HDL-C up to 10%) and Pitavastatin (increases HDL-C up to 29%). Increasing HDL-C independently with the use of four new drugs has been shown to increase cardiovascular and total mortality. Statin intolerance is a major clinical problem, including myalgia, myositis, hepatotoxicity, statin-dependent diabetes mellitus and rhabdomyolysis. Many of the mechanisms underlying its occurrence have been studied. It is the Achilles’ heel of statin therapy, frequently resulting in the discontinuation of statins. According to the latest evidence, the established statin intolerance is close to 3–5% [5].

Plant-based foods rich in phenolic phytochemicals are among the promising strategies counteracting age-related changes in lipid profile [6,7,8]. Polyphenolic compounds are found in all plants and are quantitatively a substantial type of antioxidants consumed with food. Research interest in these compounds is immense due to the numerous therapeutic effects they exert, mainly in conditions related to oxidative stress [9]. The fruits of black chokeberry, *Aronia melanocarpa* (AM), are among the richest sources of polyphenols in the plant kingdom. Polyphenols, especially antocyanins and proanthocyanidins, are the main group of biologically active components in AM berries [9,10]. They are the main contributors to the antioxidative properties of the plant. Apart from polyphenols, black chokeberry is a source of glucose, fructose, sucrose, sorbitol and pectin. Analyses have shown that AM fruit contains relatively large amounts of K and Zn, along with certain amounts of Na, Ca, Mg and Fe. Apart from minerals, other substances have been found, such as B1, B2, B6, and C vitamins, niacin, pantothenic acid, folic acid, alpha and beta tocopherol and carotenoids. Beta-sitosterol and campesterol have been identified among the triterpenes. The organic acids contained in the fruit include citric, malic, shikimic and ascorbic acid [9,11]. The antioxidative potential of AM has been confirmed by numerous in vitro studies and in vivo models, where it has been frequently associated with other medicinal properties. Other proven effects are the antihypertensive, lipid-lowering and anti-inflammatory effects of black chokeberry [12,13,14]. In the literature, no adverse or toxic effects of AM juice, fruit and extracts have been reported [9].

Research has found lipid-regulating properties in AM; however, most studies use different disease models, and evidence of the effect of black chokeberry juice or extracts in models of healthy aging animals is scarce. In a previous study of ours, the supplementation of aging rats with pure black chokeberry juice ad libitum, at a dose of 64 mL/kg per day, improved their lipid profile by lowering TC and LDL, as compared to the adult controls. However, high content of fructose, glucose and sorbitol in the juice resulted in a significant increase in the body weight of the animals under study. In order to avoid such an effect and look for the possibility of applying AM products in functional nutrition, we decided to conduct a new experiment with a modified fixed dose of 10 mL/kg, and variants including enrichment with pectin and herbal extracts [15].

A number of fruits (apples, citruses, quinces, etc.) contain pectin, which has lipid-lowering properties. Pectin normalized body weight in rats following a lipid-rich diet, along with dyslipidemia, hyperglycemia, hyperinsulinemia, metabolic endotoxemia and systemic inflammation [16]. Aside from this, various herbs used in traditional medicine have been found to exert a beneficial effect on lipid levels [17,18,19]. Elderflowers (*Sambucus nigra*) and rose hips (*Rosa canina*) are known to be rich in polyphenolic compounds belonging to different classes and exhibit a biological activity associated with lipid metabolism [20,21]. At present, we are tempted to apply substances other than drugs in controlling HDL-C, but evidence in the literature is insufficient and controversial [22,23]. Regulating dyslipidemia with functional beverages rich in phenolic phytochemicals is among the promising management strategies of age-related dyslipidemia [24]. Therefore, the aim of the present study was to investigate the effect of three AM-based functional beverages on the anthropometric parameters and lipid profile of healthy adult rats.

## 2. Materials and Methods

### 2.1. Plant Materials

*Aronia melanocarpa* berries were supplied by the licensed farmer Todor Petkov (Kazanlak, Stara Zagora district, Bulgaria) in the stage of full maturity in August 2017. Fresh fruits were packed in polyethylene bags, frozen immediately and stored at −18 °C until the time of juice production. To prepare the beverage, 5 kg of frozen fruit were defrosted at room temperature and homogenized in a laboratory blender.

*Rosa canina* (rose hip) fruits were harvested from the Rhodope Mountains in the autumn of 2016, frozen at −18 °C and freeze-dried for 96 h in Alpha 1–4 LDplus laboratory freeze dryer. After that, dried rose hips were deseeded and stored in paper bags at room temperature until the time of production of the functional beverage.

Dried elder (*Sambucus nigra*) flowers were purchased from a local pharmacy in Plovdiv, Bulgaria, ground to a fine powder in a laboratory mill and stored in paper bags prior to preparation of the extract.

### 2.2. Preparation and Characterization of Functional Beverages

#### 2.2.1. Preparation of Pure Aronia (A) Extract

To prepare the aronia extract, 400 g of fruit homogenate were mixed with 600 mL ultrapure water, transferred to a brown-glass bottle and incubated in a thermostatic water bath shaker (NUVE, Asagi Ovecler Ankara, Turkey) for 1 h, at 60 °C. After this, the mixture was filtered through a cheese cloth and the liquid phase was centrifuged (20 min, 6200× *g*) in a Megafuge 1.0R bench top centrifuge (Heraerus Instruments, Hanau, Germany).

#### 2.2.2. Preparation of Aronia and Pectin (A + P) Beverage

To prepare the aronia beverage combined with pectin, 10 g apple pectin was dissolved in 1 L of aronia extract obtained by the procedure described in Section 2.2.1.

#### 2.2.3. Preparation of Aronia with Herbs (A + H) Beverage

For the preparation of the aronia beverage with herbs, 360 g of aronia homogenate were mixed with 40 g rosehips husks (without seeds), 20 g elder flower powder and 600 mL ultrapure water. The mixture was transferred to a brown-glass bottle and incubated in a thermostatic water bath shaker (NUVE, Asagi Ovecler Ankara, Turkey) for 1 h, at 60 °C. After this, the mixture was filtered through a cheese cloth and the liquid phase was centrifuged (20 min, 6200× *g*) in a Megafuge 1.0R bench top centrifuge (Heraerus Instruments, Hanau, Germany). 

#### 2.2.4. Determination of Total Polyphenols

Total polyphenols were determined by the method of Singleton and Rossi (1965), using the Folin–Ciocalteu reagent. Gallic acid was used for the calibration curve and the results were expressed as gallic acid equivalents (GAE) per liter of beverage.

#### 2.2.5. Antioxidant Activity by Oxygen Radical Absorbance Capacity (ORAC) Assay

ORAC was measured by the method of Ou et al. (2001). ORAC analyses were carried out on a FLUOstar OPTIMA plate reader (BMG Labtech, Ortenberg, Germany) with an excitation wavelength of 485 nm and emission wavelength of 520 nm. The results were expressed as μmol of Trolox equivalents per liter (μmol TE/L).

Polyphenol content and ORAC antioxidant activity of the AM-based functional beverages under study are shown in Table 1.

### 2.3. Experimental Animals

The present study included 40 male Wistar rats. The animals were bred in the vivarium of the Medical University, Plovdiv under standard laboratory conditions (housed in polypropylene cages in a controlled environment under the following conditions: a temperature of 22 ± 3 °C, a 12-h light/dark cycle, and relative humidity of 60 ± 5%). Taking into account the hormonal influences in females, we used only male rats. The animals were divided into 5 groups (*n* = 8). Eight two-month-old rats and eight 13-month-old rats were designated as young controls (YC) and adult controls (AC), respectively. Three groups were given orally polyphenol-rich drinks diluted with drinking water in the course of 3 months, the supplemented animals being the same age as the adult controls. The groups were as follows: (A) 100% AM extract in a dose of 10 mL/kg; (A + P)—AM extract enriched with 1% pectin in a dose of 10 mL/kg; (A + H)—AM extract enriched with a herbal mixture (rosehip fruit and elderflowers) in a dose of 10 mL/kg. The rats were kept in compliance with all the experimental procedures recommended by the European Commission for protection and welfare of laboratory animals. The experimental protocol was approved by the Committee on Ethical Treatment of Animals of the Bulgarian Agency for Food Safety (No. 193/2018). All animals received humane care in compliance with the “Principles of laboratory animal care” formulated by the National Society for Medical Research and the “Guide for the care and use of laboratory animals” prepared by the National Institute of Health (NIH publication No. 86-23, revised 1996).

### 2.4. Anthropometrical Measurements

Body measurements were taken at the end of the experiment—body weight (g), body length (cm) and heart weight (g). Body mass index (body weight (g)/length^2^ (cm^2^)) and heart weight index (heart weight (g)/body weight (g) × 100) were calculated.

### 2.5. Biochemical Measurements

At the end of the experiment, all animals were anesthetized with i.m. Ketamin (90 mg∙kg) + Xilazine (10 mg∙kg) and then decapitated. Immediately after decapitation, blood was collected, placed in a centrifuge tube and allowed to clot, so that serum could be obtained. The latter was removed by centrifugation at 1400× *g* for 10 min. Serum triglycerides (TG, mmol/L) and total cholesterol (TC, mmol/L) were determined by the enzyme-colorimetric method. Serum levels of high-density lipoprotein cholesterol (HDL-C, mmol/L) and low-density lipoprotein cholesterol (LDL-C, mmol/L) were determined by the direct enzyme-colorimetric method (Beckman Coulter AU 480 chemistry analyzer). The atherogenic indices were determined as TC-C/HDL-C ratio and LDL-C/HDL-C ratio.

### 2.6. Statistical Analysis

The data obtained were processed statistically using SPSS 21. The results are presented as mean ± standard error of mean (SEM). Paired sample *t*-test, independent sample *t*-test and one-way ANOVA followed by Tukey’s test were used for the parametric analysis at a normal distribution level. Wilcoxon signed rank test and Mann–Whitney U tests were used for the non-parametric analysis. *p* < 0.05 was considered as statistically significant.

## 3. Results

### 3.1. Anthropometric Parameters

#### 3.1.1. Somatometric Parameters

At the end of the experiment, body weight and body mass index were increased in the adult animals (both control and supplemented groups), as compared to the young controls (*p* < 0.05), which is a natural occurrence in the aging process. Body weight and body mass index in the adult-supplemented animals did not change significantly, as compared to the adult controls (Figure 1A,B).

#### 3.1.2. Organometric Parameters

The analysis of heart weight showed an increase in the adult animals (both control and supplemented groups), as compared to the young group (*p* < 0.05). For example, YC (0.515 g) vs. AC (1.120 g), (*p* < 0.05), YC (0.515 g) vs. A (1.160 g), (*p* < 0.05). No significant difference was found in these parameters when comparing individuals from the supplemented adult groups (Figure 2A,B).

### 3.2. Lipidogram

#### 3.2.1. Serum Lipids and Atherogenic Indexes—Comparison between Control Groups

The results showed significant differences in the lipid parameters of the young and adult controls, confirming the spontaneous age-related dyslipidemia. Triglycerides (0.67 mmol·L^−1^) and HDL-C (1.55 mmol·L^−1^) of YC were significantly different (*p* < 0.05) in comparison to the values found in AC—1.25 mmol·L^−1^ and 1.09 mmol·L^−1^, respectively—resulting in a significantly lower (*p* < 0.05) TC/HDL index in YC (1.29), as compared to AC (1.71) (Figure 3B,D and Figure 4A). However, no significant differences were observed in TC (2.01 mmol·L^−1^ vs. 1.85 mmol·L^−1^) and LDL-cholesterol (0.34 mmol/L vs. 0.35 mmol/L) levels in the young and adult controls (Figure 3A–D).

#### 3.2.2. Serum Lipids and Atherogenic Indexes—Comparison between Supplemented and Control Groups

Supplementation with functional drinks led to certain statistically significant changes in the lipid profiles of the supplemented animals, as compared to the adult controls. For instance, the HDL-C values observed in the A + P and A + H groups (1.49 mmol/L and 1.61 mmol/L, respectively) were higher than that of the AC group (1.09 mmol/L), (*p* < 0.05), whereas the atherogenic TC/HDL index in the A + P and A + H groups (1.49 and 1.46, respectively) was significantly lower than the one found in AC (1.71), (*p* < 0.05) (Figure 3A–D). In addition, the atherogenic index LDL/HDL was improved significantly in the A + P and A + H groups, as compared to the adult controls (Figure 4A and 4B).

## 4. Discussion

### 4.1. Somatometry

From the results, it is evident that body weight alterations in the animals increase with age—a fact that has been found by other authors as well. Body weight and body mass index change significantly with the aging process, which is due to an increase in the body adipose tissue, on the one hand, and to changes in the lean mass, on the other [25,26,27]. It is also clear that AM supplementation in a dose of 10 mL·kg^−1^ does not lead to significant body weight alterations. These results showed that supplementation with functional beverages did not have an effect on the body mass index—the criterion for normal body structure. The significant differences between the weight index of the internal organs of the animals from the young and adult groups (supplemented and not supplemented) are an indication of the age-related atrophy of these organs. The absence of significant differences between the weight indices of the supplemented animals and the adult controls suggests that the weight of an organ changes in correlation to the overall body weight as a result of supplementation, which has been described by other authors [28]. Furthermore, the supplementation did not significantly affect the somatometric parameters; thus, they cannot be associated with changes in the lipid profile.

### 4.2. Age-Related Dyslipidemia

The occurrence of age-related dyslipidemia in healthy middle-aged rats was confirmed by the significant differences in the lipid profile, as compared to the young group (Figure 3 A–D). The results obtained in the current study correspond to the evidence presented by other authors [2]. It is a proven fact that total cholesterol and LDL plasma levels increase as part of the normal process of aging, while HDL-C level decreases with age [29,30]. The potential mechanisms underlying the age-related disturbance in the lipoprotein metabolism of humans and animals, which have not been fully clarified yet, are associated with changes in the liver sinusoidal endothelium, postprandial lipidemia, insulin resistance induced by free fatty acids, as well as changes in growth hormone, androgens in men and the expression and activity of peroxisome proliferator-activated receptor (PPAR) [2].

### 4.3. Lipid-Lowering Properties of Aronia

The current study involved healthy normally aging animals and followed the natural development of age-related dyslipidemia. Studies in animals and humans have shown that AM berries effectively modulate lipid metabolism [31,32,33]. In other experiments involving spontaneous or induced hyperlipidemias, the lipid-lowering effects of black chokeberry are more marked, as compared to our experiment. Valcheva-Kuzmanova et al. have reported that AM juice lowers diet-induced increase in serum TC, LDL-C and TG, an effect resulting from the large amount of polyphenols contained in the juice [29,30,31]. The possible underlying mechanisms of the lipid-lowering properties of flavonoids are as follows: cholesterol uptake suppression (by silymarin and tea catechins); improved lipoprotein catabolism (by cyanides); increased bile efflux, elimination of bile cholesterol and bile acids (proven for narginin) [30]. Quercetin, a flavonoid contained in black chokeberry, has an inhibiting effect on the enzymes that take part in the process of cholesterol synthesis and esterification [31].

Research on the mechanisms of AM lipid-lowering effect has found that black chokeberry extracts exert an effect on the expression of genes regulating cholesterol synthesis, binding and elimination in a dose-dependent manner in humans. The data obtained by Kim et al. suggest that the hypolipidemic effects of AM extract are likely to be due to an increased apical efflux of LDL-derived cholesterol and to decreased chylomicron formation in the intestines [34].

A particularly interesting finding of the study is the increase in the HDL-C levels in the groups supplemented with combined beverages—pectin-enriched AM and AM enriched with a herbal mixture. Such an effect on HDL-C is in the focus of attention of present-day research. In epidemiological studies, HDL-C is considered to be a risk marker for cardiovascular disease, so inducing elevation of HDL-C levels is a controversial issue. Since many of the drugs influencing HDL-C levels either exhibit serious side effects, or do not lower the risk of developing cardiovascular diseases, there is an active search for alternative means other than medications [35]. An experiment involving administration of a cyanidin-3-*O*-β-glucoside to mice by an oral gavage (50 mg/kg body weight) has found that the substance markedly increased serum HDL-C and apo A-1 level, as well as reduced atherosclerosis [36]. On the other hand, a diet-induced obesity model in rats has proven that supplementation with two polyphenols—resveratrol and quercetin—leads to a significant lowering of TC, TG and LDL-C levels, without affecting HDL-C [37]. In an experiment involving a hyperlipidemic model of rats, the supplementation of *Fragaria nilgerrensis* Schlecht. and *Centella asiatica* (L.) Urban in combination significantly decreased the levels of TG, TC, LDL-C, apolipoprotein B and hepatic malondialdehyde, but increased serum HDL-C, apolipoprotein A1 and hepatic superoxide dismutase [18]. A compound (polydatin) isolated from *Polygonum cuspidatum* Sieb., a widely used herb in traditional Chinese medicine, has been found to influence the lipid and glucose metabolism in mice, significantly lowering serum TG and LDL-C levels, simultaneously increasing HDL-C levels, as compared to the control mice [17]. In our study, pectin-enriched AM juice exerted a potentiated effect on the lipid profile. This is the first time when such a combination was used in the natural process of aging of rats. Various studies have proven the lipid-lowering properties of pectin. An experiment involving a 4-week administration of pectin from various sources to people with moderate hypercholesterolemia revealed its lipid-lowering effect. The levels of homocysteine and C-reactive protein were not affected [38]. Studying the effects of pectin on apoE −/− mice kept on a high lipid diet has shown a decrease in the total and LDL cholesterol levels, along with a reduction in atherosclerotic lesions [39]. A 90-day experiment with pectin supplementation in mice has proven lowering of TC, LDL-C and TG serum levels, liver malondialdehyde levels and inflammation, as well as a reduction in body weight in hypercholesterolemic animals. The lipid-lowering effects of pectin are associated with its role in the excretion of bile acids and the reduction in their intestinal reabsorption, increase in the viscosity of the intestinal content, decrease in fatty acid absorption and a possible alteration in the activity of digestive enzymes [40].

The results from the present research showed that the administration of AM juice in combination with a herbal mixture have a potentiating effect on HDL-C. This is likely to be due to the rich polyphenolic content and the synergism with the other biologically active substances. The elderflowers have a high polyphenolic content, of which phenolic acids, quercetin, kaempferol, catechin, epicatechin and narginin constitute a major part. They can affect disease processes by counteracting oxidative stress, leading to a positive effect on blood pressure and blood sugar levels, stimulating the immune system, increasing plasma enzymatic antioxidant activity and lowering uric acid [20]. Rose hip has rich polyphenolic content and an abundance of vitamins C, E, and B and carotenoids, which exhibit a synergistic antioxidative effect. In recent years, research has revealed the potential of its medicinal properties in a number of conditions, such as hyperlipidemia, hepatotoxicity, inflammatory processes, renal impairment, arthritis, obesity and cancer [21]. A potential mechanism underlying the positive effect on carbohydrate and lipid metabolism is the activation of adenosine monophosphate-activated protein kinase (AMPK) in hepatic cells [41]. Millar et al. have revealed the mechanisms underlying the effect of flavonoids on HDL-C levels, emphasizing their role in reverse cholesterol transport, high-density lipoprotein metabolism and function. Having in mind that inflammation participates in inducing HDL-C particle dysfunction, flavonoids can improve HDL-C function through a reduction in oxidative stress and inflammation [42].

### 4.4. Clinical Aspects

The optimal balance between “bad” (LDL-C) and “good” (HDL-C) in the atherogenic process is part of the history of medicine. Numerous epidemiologic findings have confirmed the fact that low HDL-C levels are associated with high cardiovascular mortality [3,4,43]. On the other hand, the antiatherogenic effects of HDL-C have been proven—it participates in reverse cholesterol transport, exhibits anti-inflammatory, anti-apoptotic, anti-thrombogenic effects, and increases NO levels [44,45,46]. Modulation of HDL-C has proved to be quite a challenge and, at present, it remains an unresolved therapeutic problem [22,23,47]. Research in the field of HDL-C physiology and pathophysiology is of great interest and focuses on establishing new treatment strategies. An individual increase in HDL-C by medications is not accepted at present, except for statins, which elevate it up to 10–15% (Rosuvastatin, Pitavastatin). Increasing HDL-C levels by means other than drugs is of significant clinical importance for medical practice. Body weight reduction, diet, continuous aerobic training and cessation of smoking are important but insufficient to optimize HDL-C levels. The results of the present study convincingly demonstrated that *Aronia melanocarpa* in combination with pectin or a herbal mixture has the potential to be used not only for prophylaxis, but also as an adjunct functional food in the diet for the purposes of managing dyslipidemia.

The results obtained in the present experimental investigation regarding the significant elevation of HDL cholesterol following supplementation with black chokeberry extract require further clinical studies involving patients. Furthermore, such an elevation in HDL-C could have an extensive clinical application under the conditions of the COVID-19 pandemic. There is evidence that HDL-C plays an important part in the normal functioning of the human immune system [48]. Recent data reported in the literature have shown a substantial correlation between the low levels of HDL-C and the risk of developing COVID-19 infection, as well as the severity of its clinical course [49,50]. All the specialized publications regarding this issue showed that people infected with coronavirus had lowered HDL serum levels. Experts advise paying special attention to patients with compromised HDL levels, since this is likely to result in other pathologic complications in the course of a COVID-19 infection. The experimental model of HDL-C modulation by black chokeberry extract became the basis of another study of ours involving patients with COVID-19. This study is being carried out at present with the aim of investigating the effect of AM extract in patients with low baseline levels of HDL cholesterol on the risk of developing COVID-19 infection and the severity of its course.

## 5. Conclusions

The results of the present study indicate that nutrition-relevant doses of functional beverages based on *Aronia melanocarpa* could counteract spontaneous age-related dyslipidemia in healthy adult rats by modulating serum HDL-cholesterol levels. The addition of pectin or herbs further increases *Aronia melanocarpa* lipid-lowering effect. More specifically, the combination of aronia with rosehip and elderflower extracts had the most pronounced HDL-cholesterol increasing effect. Therefore, the functional beverages studied can be used as a potential means for prophylaxis and as an adjunct functional food in the diet for the purposes of managing dyslipidemia.

## Figures and Tables

**Figure 1 foods-10-01641-f001:**
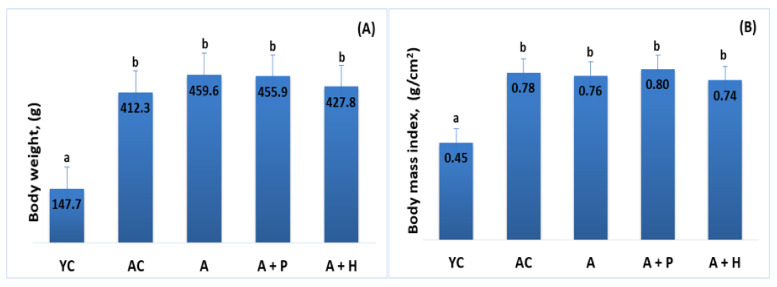
(**A**) Body weight and (**B**) body mass index of the animals from the experimental groups. The results are presented as mean values ± SEM. Different small letters indicate statistically significant differences (*p* < 0.05): a and b = YC vs. AC; YC vs. A; YC vs. A + P; YC vs. A + H.

**Figure 2 foods-10-01641-f002:**
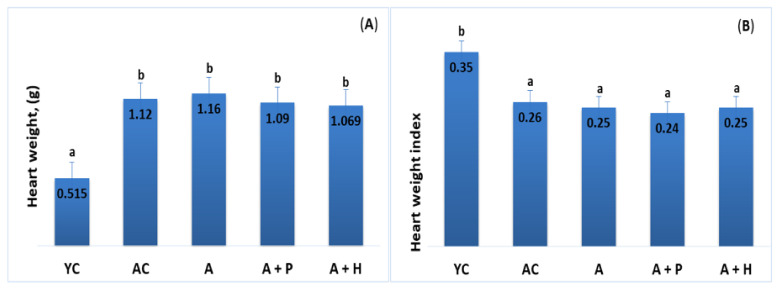
(**A**) Heart weight and (**B**) heart weight index of the animals from the experimental groups. The results are presented as mean values ± SEM. Different small letters indicate statistically significant differences (*p* < 0.05): a and b = YC vs. AC; YC vs. A; YC vs. A + P; YC vs. A + H.

**Figure 3 foods-10-01641-f003:**
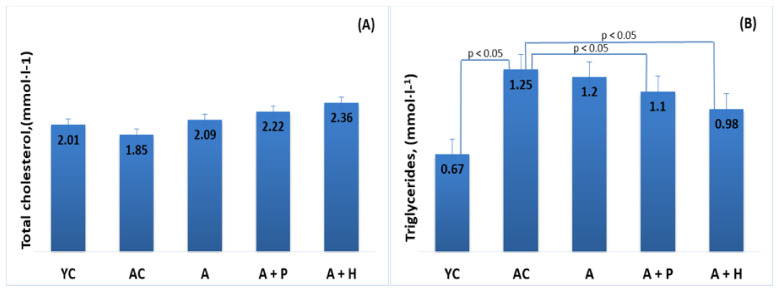
Serum (**A**) total cholesterol, (**B**) triglycerides, (**C**) LDL-cholesterol and (**D**) HDL-cholesterol of the animals from the experimental groups. The results are presented as mean values ± SEM.

**Figure 4 foods-10-01641-f004:**
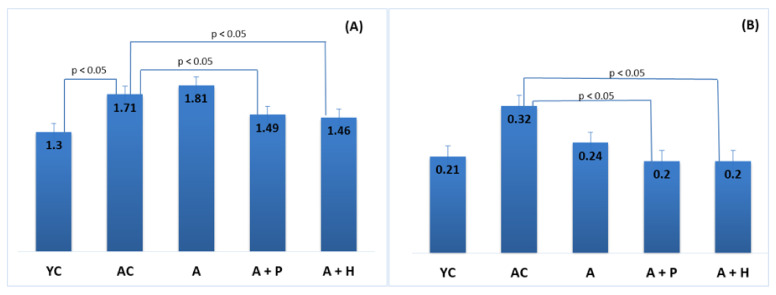
Atherogenic indices: (**A**) TC/HDL ratio and (**B**) LDL/HDL ratio.

**Table 1 foods-10-01641-t001:** Polyphenol content and ORAC antioxidant activity of aronia-based functional beverages.

Sample	Polyphenols, mg GAE/L	ORAC, µmol TE/L
A	4772 ± 256	55,307 ± 2474
A + P	4044 ± 184	52,007 ± 1704
A + H	3000 ± 311	40,900 ± 2836

## Data Availability

The data presented in this study are available on request from the corresponding author.

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
