# Peer review of "Black Chokeberry (Aronia melanocarpa) Functional Beverages Increase HDL-Cholesterol Levels in Aging Rats"

_foods, 2021, doi:10.3390/foods10071641_

Round 1
Reviewer 1 Report
The manuscript entitled: “Black Chokeberry (Aronia melanocarpa) Functional Beverages Increase HDL-Cholesterol Levels in Aging Rats” is well written.
The authors investigated the effect of three AM-based functional beverages on the anthropometric parameters and lipid profile of healthy adult rats and the results were good. The functional beverages of present work can be used in the diet for the purposes of managing dyslipidemia.
The results and conclusions are clear.
Line 135-138: the sentence would seem incomplete or unclear.
Table 1: in line 142 you have written that results were expressed as gallic acid equivalents (GAE) per liter beverage (in table 1 it wasn’t reported).
Lines 269-and 322: it is better to write “This work, present research….” rather than “Our work/results”.
Author Response
We would like to thank the reviewer for valuable comments aiming the improvement of our manuscript. All corrections in the manuscript are marked in yellow. Detailed responses to reviewer's comments and remarks are given bellow:
Line 135-138: the sentence would seem incomplete or unclear.
Answer: The sentence was revised.
Table 1: in line 142 you have written that results were expressed as gallic acid equivalents (GAE) per liter beverage (in table 1 it wasn’t reported).
Answer: We added in table 1 that polyphenol content is expressed in GAE/l.
Lines 269-and 322: it is better to write “This work, present research….” rather than “Our work/results”.
Answer: We agree with reviewer's remark and corresponding changes were made in the text.
Reviewer 2 Report
Dear autors
The described studies are very current and fit into the current research trends related to the prevention of, among others, diet-related diseases. Age and sex are physiological factors that strongly affect plasma lipid levels in a number of species. Dyslipidemia is known to be a risk factor for developing insulin resistance, endothelial dysfunction, hypertension and, most of all, cardiovascular disease. HDL-C is one of the key factors determining cardiovascular risk.
The work was very well planned and carried out. The selection of research methods does not raise any reservations. The obtained results were processed correctly, presented graphically and statistically analyzed.
The conclusions were drawn on the basis of the obtained results, they are correct, but too laconic in the opinion of the reviewer. The conclusions could clearly indicate which specific variant would be indicated in the treatment of dyslipidemia.
Author Response
We thank the reviewer for the comments. We revised the conclusion in order to show which functional beverage has the most pronounced effect on HDL cholesterol. The following text was added to the Conclusion: "More specifically, the combination of aronia with rosehip and elderflower extracts had the most pronounced HDL-cholesterol increasing effect."